# The Effects of Fermenting *Psophocarpus tetragonolobus* Tubers with *Candida tropicalis* KKU20 as a Soybean Meal Replacement Using an In Vitro Gas Technique

**DOI:** 10.3390/ani15162328

**Published:** 2025-08-08

**Authors:** Thiraphat Surakhai, Chanon Suntara, Pachara Srichompoo, Metha Wanapat, Sompong Chankaew, Anusorn Cherdthong

**Affiliations:** 1Tropical Feed Resources Research and Development Center (TROFREC), Department of Animal Science, Faculty of Agriculture, Khon Kaen University, Khon Kaen 40002, Thailand; thiraphat.su@kkumail.com (T.S.); chansun@kku.ac.th (C.S.); pachara.s@kkumail.com (P.S.); metha@kku.ac.th (M.W.); 2Department of Agronomy, Faculty of Agriculture, Khon Kaen University, Khon Kaen 40002, Thailand; somchan@kku.ac.th

**Keywords:** fermentation, plant root, ruminant, single cell protein, sustainability

## Abstract

Winged bean tuber is a local, underused crop with low crude protein (CP) content, limiting its use in ruminant diets. This study evaluated whether fermentation with *Candida tropicalis* KKU20 could improve its nutritional value and suitability as a partial replacement for soybean meal. Using an in vitro gas production technique, effects on rumen fermentation and degradability were assessed. Fermentation increased CP and improved nutrient utilization when 33% of soybean meal was replaced, particularly in high-concentrate diets. Higher inclusion levels reduced in vitro dry matter degradability. These results suggest that fermented winged bean tuber can serve as a sustainable and nutritionally viable protein source for ruminants.

## 1. Introduction

The rising global population has increased the demand for animal protein, necessitating sustainable and cost-effective protein sources for livestock. In livestock systems, protein sources represent one of the most costly and scarce elements in feed formulation [1]. Soybean meal (SBM) is the preferred protein source in animal feeds due to its high crude protein (CP) content and digestibility [1]. Its extensive utilization is attributed to its elevated CP concentration and excellent nutritional profile [2]. However, developing countries often face challenges in securing adequate SBM supplies, leading to higher costs and periodic shortages [3]. This situation highlights the need for economically viable and sustainable protein alternatives, as reliance on imported protein sources becomes increasingly unsustainable [4].

In this context, tropical legumes such as winged bean (*Psophocarpus tetragonolobus*) have emerged as promising alternatives due to their adaptability to hot, humid climates and favorable nutritional profiles [5]. The tubers of winged bean are especially noteworthy, containing 20–25% CP, 34–40% carbohydrates, and low levels of crude fat, with gross energy values ranging from 15,810 to 16,241 J/g [6]. Despite their potential, current production of winged bean tubers remains limited and localized, with no comprehensive national or global production data available. Under typical tropical conditions, winged bean tuber yields range from approximately 15.2 to 15.5 t/ha, which are slightly lower than those of cassava. However, these yields remain favorable for application in ruminant feeding systems. Recent breeding initiatives in Thailand, as reported by Rakvong et al. [6], have demonstrated improved productivity. Notably, experimental lines W018 and W099 produced 22.4 and 19.3 t/ha, respectively—surpassing the yield of the widely cultivated Ratchaburi variety. These advancements indicate the potential for expanded utilization and future integration into ruminant feed production. However, the crude protein content of winged bean tubers is still lower than that of SBM, which limits their direct substitution.

Fermentation is an effective biotechnological method for improving the nutritional quality of agricultural by-products, particularly by increasing protein content, reducing fiber levels, and enhancing digestibility [7,8]. Yeast fermentation, in particular, has been shown to increase protein content, reduce fiber levels, and improve nutrient digestibility [9]. While *Saccharomyces cerevisiae* is commonly used in animal feed fermentation, *Candida tropicalis* KKU20 was selected for its Crabtree-negative metabolism, which enables efficient glucose utilization without ethanol accumulation. Unlike *S. cerevisiae*, which channels excess glucose into ethanol, *C. tropicalis* KKU20 prioritizes biomass synthesis, leading to greater microbial protein yield and enhanced substrate utilization [10]. This property improves microbial efficiency in the rumen, potentially increasing protein availability for the host animal.

Additionally, fermentation with *C. tropicalis* facilitates fiber degradation, particularly reducing neutral detergent fiber (NDF) and acid detergent fiber (ADF), thereby improving overall feed digestibility [11]. Compared to *S. cerevisiae*, *C. tropicalis* KKU20 produces a wider array of hydrolytic enzymes, such as cellulases and proteases, which enhance fiber breakdown and protein solubilization in the rumen [10]. Furthermore, its fermentation process reduces anti-nutritional factors, including tannins and phytates, which otherwise hinder nutrient bioavailability in plant-based feeds [12,13]. These attributes position *C. tropicalis* as a promising yeast strain for improving ruminant feed quality and optimizing fermentation efficiency.

While *C. tropicalis* has been explored in fermentation processes, its role in enhancing ruminant feed ingredients remains underexplored. This study hypothesizes that fermenting winged bean tuber with *C. tropicalis* KKU20 will increase its protein content, improve fiber degradability, and enhance ruminal fermentation efficiency compared to unfermented WBT. The objective is to evaluate the effects of replacing SBM with yeast-fermented WBT on ruminal fermentation, gas kinetics, and in vitro degradability, determining the optimal inclusion level for ruminant nutrition.

## 2. Materials and Methods

### 2.1. Animal Ethics

All experimental procedures in this study were reviewed and approved by the Ethical Committee of Khon Kaen University, in accordance with the guidelines set by the National Research Council of Thailand. The approval was granted under protocol number IACUC-KKU-49/67, issued on 17 June 2024, with additional reference No. 660201.2.11/441 (47), ensuring full compliance with animal welfare standards.

### 2.2. Study Site, Winged Bean Tuber Source, and C. tropicalis KKU20

The winged bean tubers (WBT; *Psophocarpus tetragonolobus* (L.) DC.) used in this study were sourced from the Department of Agronomy at Khon Kaen University, Thailand. Fresh WBT used in this in vitro experiment was sun-dried for 3–4 days, chopped into 3–5 cm lengths, and subjected to fermentation. The ruminal yeast strain *C. tropicalis* KKU20 (CBS 94T, U45749) was obtained from previous studies [10] and selected for its ability to enhance microbial protein synthesis and fiber degradation.

### 2.3. Yeast-Fermented Winged Bean Tubers

*C. tropicalis* KKU20 yeast powder (1 × 10^13^ cells/g) was reactivated and cultured in a liquid medium composed of 250 g of molasses (sourced from Khon Kaen Dairy Cooperative Co., Ltd., Khon Kaen, Thailand) and 10 g of urea dissolved in one liter of water. The fermentation process was conducted at 30 °C for seven days under anaerobic conditions, with the pH adjusted to 3.5 using formic acid to optimize yeast metabolism and minimize bacterial contamination. Throughout the fermentation period, samples were monitored for microbial stability and pH fluctuations to maintain consistent fermentation quality. To promote yeast growth and aerobic respiration, the culture medium was continuously aerated for 60 h using an electromagnetic air compressor (HAILEA ACO-318, Sagar Aquarium, Gujarat, India), with oxygen supplied throughout the process. At the end of incubation, the yeast concentration reached 1.15 × 10^11^ cells/mL.

The aeration step using an oxygen pump was employed during the yeast propagation phase to support cell growth, but anaerobic conditions were maintained during the main fermentation. The medium was then mixed with WBT at a 1:1 ratio (medium: WBT). The desiccated WBT was thoroughly mixed with the yeast solution and evaluated for fermentation quality. Thirty kilograms of the yeast–WBT mixture was manually packed into plastic bags (60 × 110 cm, Silver Bag Limited Partnership Co., Ltd., Nakhon Ratchasima, Thailand), vacuum-sealed, and stored at ambient temperature for seven days. After fermentation, the nutritional composition of all fermented WBT samples was analyzed [14].

### 2.4. Experimental Design and Dietary Treatments

A 3 × 4 factorial arrangement in a completely randomized design was employed. The experimental design included three roughage-to-concentrate (R:C) ratios (60:40, 50:50, and 40:60) and four levels of soybean meal replacement with YFWBT (0%, 33%, 66%, and 100%) to evaluate their interactive effects on rumen fermentation and degradability. Rice straw was used as the roughage source.

YFWBT samples were oven-dried at 60 °C for 48 h using a forced-air system until moisture content stabilized. The dried material was then ground with a Wiley mill (Arthur H. Thomas Co., Philadelphia, PA, USA) fitted with a 1 mm mesh screen in preparation for chemical composition analysis.

The chemical composition of the YFWBT was analyzed for dry matter (DM; method ID 967.03), ash (ID 492.05), and ether extract (EE; ID 445.08). Crude protein was measured using a LECO FP-828 P combustion analyzer (LECO Corporation, St Joseph, MI, USA), with nitrogen content converted to protein by applying a factor of 6.25. Neutral detergent fiber and acid detergent fiber (ADF) concentrations were measured using the standard detergent fiber analysis technique [15]. The nutrient profiles of the concentrate mixtures and rice straw are summarized in Table 1. While urea was included to ensure isonitrogenous conditions among treatments, it is acknowledged that in vitro systems do not replicate the absorption and excretion processes occurring in vivo. Consequently, the excess availability of inorganic nitrogen in the medium may influence microbial activity and fermentation dynamics, particularly in the context of protein evaluation.

### 2.5. Animal Donors and Ruminal Inoculum Preparation

Rumen fluid was collected from two healthy male Thai-native beef cattle (average body weight 273 ± 16.0 kg) that were housed in individual pens with free access to clean water. The animals were fed a concentrate diet at 0.5% of body weight twice daily (07:00 and 16:00 h), formulated to contain 14.0% crude protein and 75.0% total digestible nutrients (TDN), comprising cassava chips, corn meal, rice bran, soybean meal, palm kernel meal, urea, salt, minerals, and vitamins. Rice straw was provided ad libitum. This feeding regimen was maintained for seven days prior to rumen fluid collection. On the day of collection, before the morning feeding, approximately 2000 mL of rumen fluid was obtained from each animal using a stomach tube connected to a vacuum pump. The tube was inserted into the mid-rumen region, and the initial portion of fluid was discarded to minimize saliva contamination. Rumen fluid from both animals was then pooled in equal proportions into pre-warmed thermos flasks (maintained at 39 °C), yielding approximately 3.5 L of mixed inoculum. The fluid was immediately filtered through eight layers of cheesecloth before being used as the microbial inoculum in the in vitro fermentation procedure. The in vitro substrate test consisted of 0.5 g of DM from the experimental diet, with rice straw as the roughage source. Roughage, concentrate, and YFWBT were ground and sieved through a 1 mm screen using a Cyclotech Mill (Tecator, Höganäs, Sweden) prior to incubation. The feed samples were then accurately weighed into 50 mL incubation bottles, which were sealed with rubber stoppers and aluminum caps. To create anaerobic conditions, the headspace of each bottle was flushed with carbon dioxide (CO_2_) to displace oxygen.

Artificial saliva was prepared according to the modified method of Cherdthong and Wanapat [16], incorporating distilled water, macro- and microminerals, and a buffer solution to simulate ruminal conditions. To create the incubation medium, the artificial saliva was mixed with rumen fluid at a ratio of 2:1 under continuous CO_2_ flushing to maintain anaerobic conditions. Each 50 mL bottle, preloaded with 0.5 g of the experimental diet, was injected with 40 mL of the rumen fluid–saliva mixture using a sterile 1.5-inch, 18-gauge needle. The sealed bottles were then incubated at 39 °C for a total of 96 h. Although the Menke in vitro gas production technique was originally developed to estimate the energy value of feedstuffs, it was used in this study to evaluate the fermentative behavior of complete diets. These diets included YFWBT as a partial substitute for soybean meal. YFWBT provides both fermentable carbohydrates and nitrogenous compounds, which can influence rumen fermentation dynamics. Therefore, the technique remains relevant for assessing overall degradability and fermentation characteristics. However, its limitations in directly measuring protein quality or nitrogen utilization are acknowledged.

### 2.6. In Vitro Gas Production and Fermentation Characteristics

The experimental design included 12 dietary treatments, each with four replicates, resulting in 48 bottles (4 bottles/treatment × 12 treatments) for in vitro gas production kinetics. To account for baseline gas accumulation, four blank bottles containing only the buffer medium were included, and their values were subtracted to calculate net gas production. For the analysis of rumen fermentation parameters—including pH, ammonia-nitrogen (NH_3_-N), and volatile fatty acids (VFAs)—a total of 96 bottles were prepared (4 bottles/treatment × 12 treatments × 2 incubation times at 24 and 48 h). An additional 96 bottles, prepared following the same treatment and time point structure, were used for determining in vitro degradability. All bottles were incubated under standardized conditions, ensuring consistency across replicates and time points to enhance the reliability of the results.

Gas production was monitored using a 10 mL glass hypodermic syringe affixed to the incubation container at a distance of 1.5 inches. Gas emission was recorded at the following time intervals: 0, 0.5, 1, 2, 3, 4, 6, 8, 9, 12, 24, 48, 72, and 96 h after incubation. Gas production kinetics were analyzed using the following model by Schofield [17]:Gas production = b [1 − exp^−c(t−L)^],
where *b* represents the asymptotic gas volume (mL/g DM) corresponding to the complete degradation of the substrate, *t* is the incubation time (hours), *c* is the fractional rate of gas production per hour, and *L* denotes the lag phase (hours) before fermentation begins.

The pH of the fermentation medium was measured at both 24 and 48 h post-inoculation using a total of 99 bottles, with 36 samples analyzed at each time point. Prior to analysis, the contents were filtered through Grade 40 cheesecloth. For ammonia–nitrogen (NH_3_-N) and volatile fatty acid (VFA) determination, previously frozen rumen fluid samples were thawed and centrifuged at 16,000× *g* for 10 min. VFA profiles were then quantified using gas chromatography, following the protocol outlined by Yamamoto-Osaki [18].

Ammonia–nitrogen concentration was determined using the colorimetric method described by Fawcett and Scott [19]. Briefly, 40 µL of the centrifuged rumen fluid was mixed with 2500 µL of phenol color reagent and 2000 µL of alkaline hypochlorite reagent in a 15 mL test tube using a vortex mixer. The mixture was incubated at 37 °C for 10 min in a shaking water bath to allow for the development of a blue-colored complex. Absorbance was then measured at 630 nm using a spectrophotometer (Evolution 160 UV-Vis Spectrophotometer, Thermo Fisher Scientific Inc., Waltham, MA, United States).

In vitro dry matter degradability (IVDMD) and in vitro organic matter degradability (IVOMD) were determined at 24 and 48 h of incubation using a modified version of the Tilley and Terry [20] method, in which only the ruminal phase was applied without the second-stage pepsin digestion.

### 2.7. Statistical Analysis

All data were analyzed using analysis of variance (ANOVA) with the General Linear Model (PROC GLM) procedure in SAS software version 9.4 [21]. The data were analyzed according to the following model:Y_ijk_ = µ + a_i_ + bj + (ab)_ij_ + ɛ_ijk_
where Y_ijk_ represents the response variable, μ is the overall mean, a_i_ represents the levels of the R:C ratio (60:40, 50:50, and 40:60; i = 1–3), b_j_ represents the levels of YFWBT inclusion (0%, 33%, 66%, and 100% of DM; j = 1–4), (ab)_ij_ represents the interaction between the two factors, and ɛ_ijk_ is the residual error.

Each bottle was treated as an independent experimental unit. Prior to ANOVA, data were tested for normality (Shapiro–Wilk test) and homogeneity of variance (Levene’s test), and all assumptions were met. Results are presented as means with their corresponding standard error of the mean (SEM). When significant differences were detected (*p* < 0.05), Tukey’s honestly significant difference (HSD) test was applied for post hoc multiple comparisons among treatment means to control type I error across the factorial design.

## 3. Results

### 3.1. Ingredient Composition and Chemical Composition

Table 1 summarizes the chemical composition of experimental feeds, including concentrate, rice straw, WBT, and YFWBT. The concentrate diets exhibited consistent CP levels across all groups, ranging from 14.10% to 14.46% DM. Urea was incorporated into the concentrate formulations to achieve the desired protein content. The CP content of WBT was 15.35%, which increased to 17.71% in YFWBT after fermentation, representing a 13.32% increase. The NDF content in WBT was 24.21%, which decreased to 19.38% in YFWBT after fermentation, reflecting a reduction of 19.95%. Similarly, the ADF content in WBT was 10.58%, which decreased to 6.60% in YFWBT, corresponding to a reduction of 37.62%. This enhancement in protein concentration and the reduction in fiber content highlight the effectiveness of the fermentation process in improving the nutritional value of WBT.

### 3.2. Kinetics of Gas Production

The gas production kinetics parameters of the tested substrates are presented in Table 2, and the cumulative gas production profiles over time are illustrated in Figure 1. Although gas production and IVDMD were measured in separate incubations, the estimated partition factor (PF) was calculated by dividing the IVDMD by the corresponding gas volume for each treatment. These PF estimates provide insight into the relative microbial fermentation efficiency among treatments. No significant interaction was detected between the R:C ratio and YFWBT level for cumulative gas production at 24, 48, or 96 h of incubation (*p* > 0.05). At 96 h, cumulative gas production remained unaffected by either the R:C ratio or YFWBT inclusion. However, at earlier incubation times (24 and 48 h), cumulative gas production increased with higher concentrate proportions. Similarly, YFWBT-supplemented groups showed greater gas production at 24 and 48 h compared to the control. Regarding gas production kinetics, the R:C ratio significantly affected both the gas production rate constant (‘c’) and the lag phase (‘L’) (*p* < 0.01), with faster gas generation observed at R:C ratios of 50:50 and 40:60. Additionally, a higher concentrate proportion extended the lag time, potentially reflecting delayed microbial adaptation. In contrast, increasing levels of YFWBT significantly influenced the asymptotic gas volume (‘b’) (*p* < 0.05).

### 3.3. In Vitro Ruminal Fermentation

Table 3 summarizes the effects of the R:C ratio and YFWBT levels on ruminal NH_3_-N concentration and pH values after 24 and 48 h of incubation. Both factors significantly influenced NH_3_-N concentration and pH (*p* < 0.01). After 24 h, NH_3_-N concentrations ranged from 7.66 to 13.76 mg/dL, increasing to 16.44–16.63 mg/dL after 48 h. Meanwhile, pH values remained relatively stable, ranging from 6.60 to 6.75 at 24 h and maintaining a similar range at 48 h. These variations highlight the interactive effects of YFWBT inclusion and the R:C ratio, suggesting that YFWBT enhances microbial activity without disrupting ruminal stability.

### 3.4. In Vitro Degradability

Table 4 shows the effects of different R:C ratios and YFWBT inclusion levels on in vitro dry matter degradability (IVDMD) and in vitro organic matter degradability (IVOMD) at 24 and 48 h of incubation. A significant interaction between the R:C ratio and YFWBT level was observed for both IVDMD and IVOMD (*p* < 0.01). Increasing the proportion of concentrate alongside YFWBT supplementation improved degradability values at both time points. After 24 h, IVDMD ranged from 41.89% to 52.67%, increasing to 52.50–64.49% at 48 h. The highest IVDMD (64.49%) was observed at an R:C ratio of 40:60 with 33% YFWBT inclusion. A similar trend was noted for IVOMD, with peak values recorded at the same combination, reaching 65.81% at 48 h.

### 3.5. In Vitro Volatile Fatty Acids

Table 5 illustrates the effects of varying R:C ratios and levels of YFWBT on VFA profiles after 24 and 48 h of in vitro incubation. Significant interaction effects (*p* < 0.05) were observed at 48 h for total VFA, propionic acid (C3), and butyric acid (C4) concentrations. The highest total VFA was found in the 40:60 group with 33% YFWBT (104.31 mM at 48 h). Both C3 (26.22%) and C4 (19.30%) also peaked at 48 h in the 40:60 group with 66% YFWBT. At 24 h, total VFA concentration was significantly affected by the R:C ratio (*p* < 0.05), with the lowest values observed in the 60:40 group, and increased with higher concentrate levels and YFWBT inclusion. For acetic acid (C2) and the acetate-to-propionate (A:P) ratio, no interaction effects were observed. Acetic acid concentration was highest in the 60:40 group (59.77% at 48 h) and declined significantly with increasing concentrate levels (*p* < 0.05). The A:P ratio also decreased significantly (*p* < 0.01) as concentrate levels increased.

## 4. Discussion

### 4.1. Ingredient Composition and Chemical Composition

In this study, the protein content of unfermented WBT was determined to be 15.35%, which closely aligns with previous findings reporting a protein content of 13.7–20% [6,22,23]. However, a slightly higher protein content of 18.98% was observed in the study by Suntara et al. [23], likely due to variations in harvest age and soil fertility management. The fermentation process significantly enhanced the chemical composition of WBT, utilizing Crabtree-negative yeast as the microbial agent and urea as an additive to optimize fermentation efficiency. This improvement in protein content is consistent with the findings of Suntara et al. [23]; however, the final CP level of 17.71% in YFWBT remains significantly lower than that of soybean meal (~44–48%). To compensate for this difference and maintain similar dietary CP content across treatments, urea was included in the concentrate formulation. Thus, the observed fermentation and IVDMD effects are due to the combined contributions of YFWBT and non-protein nitrogen from urea, rather than YFWBT alone. Similarly, Kang et al. [24] reported that supplementing cassava top silage with a 10 g/L urea solution increased its protein content from 22.0% to 32.5%, demonstrating the effectiveness of urea in improving nutritional quality. Furthermore, the reductions in NDF and ADF concentrations observed during yeast fermentation can be attributed to the enzymatic activity of cellulase, xylanase, and ligninase, which degrade fiber components [25]. Supporting this, Gunun et al. [26] found that yeast fermentation in combination with rubber seed kernel reduced NDF and ADF by 32.55% and 39.88%, respectively. These findings highlight the potential of fermentation as a strategy to enhance the nutritional profile of WBT for animal feed applications.

### 4.2. Kinetics of Gas Production

Mathematical modeling has proven effective in describing gas production dynamics during in vitro fermentation and offers insights into the kinetics of substrate degradation over time [17]. In this study, the inclusion of yeast-fermented winged bean tuber (YFWBT) influenced gas production behavior, as reflected in changes to the asymptotic gas production (‘b’) parameter. The observed increase in ‘b’ with higher levels of YFWBT suggests improved substrate fermentability, potentially due to partial fiber degradation and enhanced solubility brought about by the yeast fermentation process. This finding supports previous studies that demonstrated enhanced fiber solubilization from yeast fermentation, even in the absence of changes in fermentation rate [27,28].

Interestingly, the gas production rate constant (‘c’) remained unaffected by YFWBT inclusion, indicating that while more fermentable substrate was available, the microbial fermentation rate itself was not significantly accelerated. This observation aligns with prior reports where improvements in substrate quality through fermentation did not necessarily alter microbial kinetics [27].

The roughage-to-concentrate (R:C) ratio, however, significantly influenced gas production kinetics. Increasing the proportion of concentrate in the diet enhanced the gas production rate (‘c’), likely due to the greater availability of rapidly fermentable carbohydrates, which promote microbial activity and gas output [29]. These results are consistent with Suriyapha et al. [30], who reported increased gas production rates at higher concentrate levels. However, increasing the concentrate proportion unexpectedly prolonged the lag phase (‘L’), indicating a delayed microbial response. While higher concentrate levels typically enhance access to fermentable carbohydrates and reduce lag time, the opposite trend in this study may be due to a transient drop in pH during early fermentation, which can inhibit fibrolytic microbes. Additionally, the rumen microbial community may require time to adjust and induce enzymes for degrading starch-rich substrates. These findings highlight the complex microbial adaptation to dietary changes, even under in vitro conditions. Despite these kinetic differences, total cumulative gas production at 96 h was not significantly influenced by either the R:C ratio or the YFWBT substitution level, indicating that overall fermentation eventually reached a comparable endpoint across treatments. This aligns with Botia-Carreño et al. [31], who reported that protein-rich ingredients contribute minimally to gas volume due to limited fermentation of amino acids into gaseous products. Furthermore, as protein degradation leads to ammonia release, the resulting rise in pH can suppress CO_2_ release by promoting its dissolution as carbonic acid in the medium [16]. Nonetheless, gas production at earlier time points (24 and 48 h) was significantly greater in diets with higher concentrate proportions and YFWBT inclusion, suggesting improved fermentability of carbohydrate fractions and enhanced microbial activity during the initial stages of digestion.

### 4.3. In Vitro Ruminal Fermentation

Ruminal pH serves as a key indicator of microbial activity and fermentation balance, with optimal values typically ranging from 6.5 to 7 in healthy ruminants [32]. In this study, increasing the proportion of concentrate in the diet resulted in a significant reduction in ruminal pH, with the lowest values observed at R:C ratios of 50:50 and 40:60 without YFWBT supplementation. This decline is likely due to enhanced microbial fermentation of rapidly fermentable carbohydrates, leading to increased production of VFAs, particularly C3. The lower pH observed under higher concentrate levels may reflect a shift in fermentation pathways favoring C3 synthesis, which is commonly associated with increased starch fermentation [29]. These findings underscore the acidifying effect of high-concentrate diets and their influence on fermentation end-products [33].

The observed reduction in pH with increasing concentrate levels in this in vitro study aligns with the general trend reported in previous research. Although the absolute pH change was modest due to the buffering capacity of the medium, the direction of change reflects the expected pattern of greater acid production from fermentable carbohydrates. Ramos et al. [33] noted that high concentrate intake can lower ruminal pH and negatively affect cellulolytic bacterial activity in vivo. Similarly, Agle et al. [34] observed reduced ruminal pH with high-concentrate diets in dairy cows. In contrast, Bünemann et al. [35] reported no significant pH change, likely due to limited variation in dietary starch. While the magnitude of pH shifts in vitro is smaller, these results support the notion that both concentrate level and carbohydrate type influence fermentation dynamics and microbial environment.

At 24 h post-incubation, NH_3_-N concentrations ranged from 7.66 to 13.76 mg/dL, with higher values observed in treatments containing YFWBT. This elevation is likely due to the presence of urea incorporated into both the YFWBT and the concentrate mixtures to balance crude protein levels across treatments. The rapid hydrolysis of urea, a non-protein nitrogen (NPN) source, by microbial urease enzymes contributes to increased ammonia accumulation in the fermentation medium [36]. Additionally, differences in NH_3_-N concentration may be partially attributed to the fact that the diets with varying R:C ratios were not isonitrogenous, potentially influencing the amount of nitrogen available for microbial activity. It is important to emphasize that elevated ammonia concentrations in vitro do not necessarily reflect improved nitrogen utilization; rather, they may indicate excess nitrogen not effectively incorporated into microbial biomass [37]. This observation is in line with trends reported by Musco et al. [38], who found increased ruminal NH_3_-N levels in high-concentrate diets. However, due to fundamental differences between in vitro and in vivo systems—such as the absence of ammonia absorption through the rumen epithelium and systemic nitrogen recycling—these results should be interpreted with caution and not assumed to directly reflect nitrogen utilization efficiency in live animals. Future in vivo studies are needed to confirm whether elevated NH_3_-N concentrations under these dietary conditions translate into efficient microbial protein synthesis or indicate nitrogen losses.

### 4.4. In Vitro Degradability

In this study, both IVDMD and IVOMD at 24 and 48 h were highest when SBM was partially replaced with YFWBT at the lowest inclusion level (33% of dietary dry matter). However, as the level of YFWBT substitution increased to 66% and 100%, a notable reduction in IVDMD and IVOMD was observed. This decline can be partially explained by the increase in NDF content, which rose by approximately 2% compared to the SBM-based diet, as YFWBT inherently contains higher levels of structural carbohydrates. Increased fiber content may reduce microbial access to fermentable substrates, thereby limiting degradation efficiency [26,27]. Moreover, the observed decrease in IVDMD and IVOMD cannot be solely attributed to fiber. It is also plausible that residual anti-nutritional compounds such as tannins and phytates present in YFWBT inhibited microbial enzymatic activity, further impairing fermentation and nutrient breakdown [3,4]. These findings support the hypothesis that both elevated fiber levels and anti-nutritional factors contribute to the reduced degradability at higher YFWBT inclusion levels [30].

Similar trends were observed by Suriyapha et al. [30], who reported that replacing SBM with yeast waste-fermented citric waste (CWYW) up to 100% reduced IVDMD. They attributed this decline to the presence of structural carbohydrates in CWYW, which hindered degradability in in vitro fermentation. The higher fiber content in both YFWBT and CWYW suggests that complex carbohydrate structures present in these materials pose challenges for microbial digestion, limiting overall degradability.

Among the treatments, 33% YFWBT inclusion yielded the highest IVDMD and IVOMD values. This improvement may be attributed to the combined effects of moderate protein enrichment and reduced fiber content, which together enhance ruminal microbial activity. Additionally, the 40:60 R:C ratio significantly improved IVDMD, likely due to the greater proportion of concentrate providing readily fermentable carbohydrates that are more efficiently utilized than fibrous roughage [35,38]. While higher concentrate levels are generally associated with increased IVDMD, this effect may not be linear and is influenced by ruminal fermentation conditions [38,39]. As previously discussed, the reduction in ruminal pH associated with higher concentrate inclusion may suppress the activity of fibrolytic microbes, potentially limiting fiber degradation. Therefore, the observed improvement in IVDMD at 33% YFWBT and a 40:60 R:C ratio is more likely due to an optimal balance between fermentable carbohydrate availability and fiber content, rather than a true synergistic effect. These observations align with findings from Chen et al. [39] and Vivares et al. [40], who reported enhanced degradability in animals fed concentrate-rich diets, although further studies are needed to determine whether these effects follow a linear or threshold response.

### 4.5. Ruminal Volatile Fatty Acid Concentration

At 24 h of incubation, diets with higher concentrate levels resulted in greater total VFA production, likely due to the increased availability of fermentable carbohydrates that stimulated microbial fermentation. In contrast, lower total VFA concentrations observed in the roughage-to-concentrate ratio of 60:40 may reflect slower degradation of fiber-rich substrates and limited energy availability for microbial growth [41]. By 48 h, the interaction between the roughage-to-concentrate ratio and the level of yeast–YFWBT inclusion became more evident. Notably, the highest total VFA and C3 concentrations were observed in the 40:60 group with 33% and 66% YFWBT inclusion. This suggests that partial substitution of soybean meal (SBM) with YFWBT promoted favorable fermentation conditions. Compared to SBM, YFWBT—due to its partial degradation during the fermentation process and the presence of residual non-protein nitrogen—may provide a more synchronized release of energy and nitrogen, thus enhancing microbial growth and fermentation efficiency. The reduced fermentation response at 100% YFWBT inclusion could be due to nutrient imbalance or excess non-protein nitrogen, potentially disrupting microbial activity and fermentation dynamics [42].

Elevated C3 concentrations in high-concentrate, moderate-YFWBT diets align with increased activity of amylolytic bacteria, which utilize starch-rich substrates to produce glucogenic VFAs [14,43]. This microbial shift is consistent with energy-directed fermentation, which benefits growth performance. Conversely, C2 concentrations were highest in the 60:40 group, consistent with the dominance of fibrolytic bacteria under high-roughage conditions [42,44].

A progressive decline in the A:P ratio with increasing concentrate and YFWBT levels reflects a metabolic transition from fiber to starch fermentation. While this may lower acetate availability for lipogenesis in dairy animals, it supports more efficient energy production via C3, particularly beneficial for growing animals [45].

Overall, the combination of a 40:60 R:C ratio and moderate YFWBT inclusion provided optimal fermentation conditions, enhancing total VFA production and favoring energy-yielding fermentation profiles without compromising rumen function.

## 5. Conclusions

Fermentation of winged bean tuber with *C. tropicalis* KKU20 improved its CP content and reduced fiber levels, enhancing its nutritional profile as a feed ingredient. However, the CP concentration of the fermented product remained lower than that of SBM. To ensure comparable dietary CP levels across treatments, urea was supplemented as a source of non-protein nitrogen. When incorporated into ruminant diets at moderate levels—specifically at 33% replacement of soybean meal in a 40:60 R:C ratio—led to improved IVDMD, IVOMD, cumulative gas production and VFA concentration. These findings highlight the potential of YFWBT as a sustainable and locally sourced alternative to soybean meal, particularly in tropical regions where feed import dependence is a concern. In contrast, complete replacement of soybean meal with YFWBT (100%) reduced feed degradability and concentration of total VFA. These findings suggest that YFWBT, when co-supplemented with non-protein nitrogen, can partially replace soybean meal in ruminant diets without compromising fermentation or nutrient availability. Further in vivo research is needed to validate these outcomes, particularly with regard to nitrogen utilization efficiency, animal performance, and economic viability under practical feeding systems.

## Figures and Tables

**Figure 1 animals-15-02328-f001:**
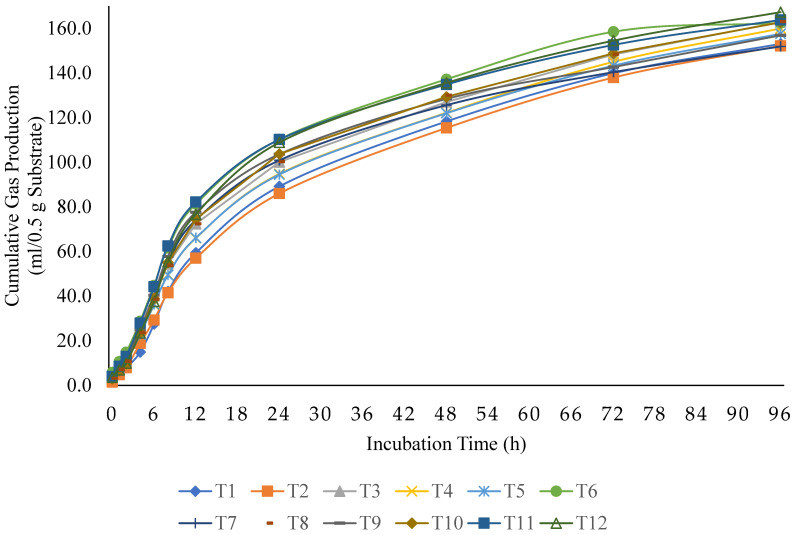
Effects of replacing soybean meal with yeast-fermented winged bean tuber on cumulative gas production over 96 h, highlighting variations across roughage-to-concentrate ratios and replacement levels [T1 = roughage-to-concentrate (R:C) ratios at 60:40 + 0% yeast-fermented winged bean tuber (YFWBT), T2 = R:C at 60:40 + 33% YFWBT, T3 = R:C at 60:40 + 66% YFWBT, T4 = R:C at 60:40 + 100% YFWBT, T5 = R:C at 50:50 + 0% YFWBT, T6 = R:C at 50:50 + 33% YFWBT, T7 = R:C at 50:50 + 66% YFWBT, T8 = R:C at 50:50 + 100% YFWBT, T9 = R:C at 40:60 + 0% YFWBT, T10 = R:C at 40:60 + 33% YFWBT, T11 = R:C at 40:60 + 66% YFWBT, T12 = R:C at 40:60 + 100% YFWBT].

**Table 1 animals-15-02328-t001:** Ingredient and chemical composition of concentrate diet and rice straw (%DM).

Item	YFWBT 0%	YFWBT 33%	YFWBT 66%	YFWBT 100%	WBT	YFWBT	Rice Straw
Ingredients, % DM							
Cassava chips	45.00	45.00	45.00	45.00			
Soybean meal	10.00	6.70	3.40	0.00			
YFWBT	0.00	3.30	6.60	10.00			
Plam kernel meal	15.80	15.50	15.30	15.10			
Rice bran	15.00	15.00	15.00	15.00			
Corn meal	12.00	12.00	12.00	12.00			
Urea	1.20	1.50	1.70	1.90			
Vitamin and mineral premix	0.50	0.50	0.50	0.50			
Salt	0.50	0.50	0.50	0.50			
Chemical composition							
Dry matter, %	93.12	93.16	93.07	93.07	35.18	51.73	96.76
Organic matter, % DM	95.60	95.68	95.87	95.96	96.53	96.15	88.74
Crude protein, % DM	14.46	14.69	14.37	14.10	15.35	17.71	3.55
Ether extract, % DM	4.89	4.83	4.79	4.58	0.32	0.54	0.93
Neutral detergent fiber, % DM	22.36	22.98	23.64	24.38	24.21	19.38	70.02
Acid detergent fiber, % DM	10.15	6.64	9.07	9.08	10.58	6.60	47.44

WBT = winged bean tuber; YFWBT = yeast-fermented winged bean tuber.

**Table 2 animals-15-02328-t002:** Gas kinetics and cumulative gas production with varying replacement levels of soybean meal by yeast-fermented winged bean tuber (YFWBT).

R:C Ratio	YFWBT (%)	Gas Kinetics			Cumulative Gas (mL/0.5 g DM Substrate)		
		b	c	L	24 h	48 h	96 h
60:40	0	101.7	0.040	0.60 ^a^	89.2	118.3	153.1
	33	101.4	0.031	0.15 ^d^	86.0	115.4	152.2
	66	103.9	0.042	0.25 ^cd^	100.1	127.0	162.9
	100	105.2	0.042	0.15 ^d^	94.8	122.2	160.0
50:50	0	102.9	0.040	0.25 ^cd^	94.5	122.0	157.6
	33	105.8	0.052	0.40 ^abcd^	110.3	137.2	162.4
	66	96.4	0.064	0.55 ^ab^	100.9	125.6	151.9
	100	105.3	0.054	0.30 ^bcd^	100.3	130.0	163.1
40:60	0	98.2	0.061	0.50 ^abc^	103.6	128.4	156.9
	33	103.6	0.053	0.50 ^abc^	103.7	129.3	162.7
	66	103.7	0.054	0.50 ^abc^	110.2	134.9	163.8
	100	107.5	0.063	0.65 ^a^	108.9	135.6	167.3
	SEM	2.09	0.003	0.08	10.23	12.44	13.58
Main effects							
	60:40	103.1	0.041 ^b^	0.29 ^b^	92.5 ^b^	120.7 ^b^	157.8
R:C Ratio	50:50	102.6	0.050 ^a^	0.38 ^b^	101.5 ^a^	128.5 ^a^	158.1
	40:60	103.2	0.052 ^a^	0.54 ^a^	106.6 ^a^	132.1 ^a^	163.0
	0	100.9 ^b^	0.040	0.45	95.8 ^b^	122.9 ^b^	155.9
YFWBT	33	103.6 ^ab^	0.041	0.35	100.0 ^a^	127.3 ^a^	159.8
	66	101.4 ^b^	0.052	0.43	103.7 ^a^	129.2 ^a^	159.5
	100	105.9 ^a^	0.042	0.37	101.3 ^a^	129.3 ^a^	163.5
Significance of main effect and interaction							
R:C Ratio		0.901	0.009	0.008	0.003	0.010	0.175
YFWBT		0.045	0.248	0.368	0.040	0.041	0.100
Interaction		0.100	0.418	0.004	0.748	0.679	0.216

^a–d^, Means with different letters in a column are significantly different at *p* < 0.05; YFWBT = yeast-fermented winged bean tuber; SEM = standard error of the mean; b = the final asymptotic gas volume corresponding to fully digested substrate (mL/g DM); c = a rate constant (units time); L = a discontinuous lag term (h).

**Table 3 animals-15-02328-t003:** Effects of replacing soybean meal with yeast-fermented winged bean tuber on ruminal pH and ammonia–nitrogen (NH_3_-N) concentration.

R:C Ratio	YFWBT (%)	pH		NH_3_-N (mg/dL)	
		24 h	48 h	24 h	48 h
60:40	0	6.75 ^a^	6.75 ^a^	12.73 ^ab^	16.56 ^abc^
	33	6.75 ^a^	6.70 ^ab^	7.66 ^d^	16.51 ^abc^
	66	6.70 ^ab^	6.70 ^ab^	10.85 ^c^	16.60 ^ab^
	100	6.70 ^ab^	6.65 ^b^	11.44 ^c^	16.50 ^abc^
50:50	0	6.60 ^c^	6.60 ^b^	11.97 ^bc^	16.64 ^a^
	33	6.70 ^ab^	6.60 ^b^	11.49 ^c^	16.51 ^abc^
	66	6.70 ^ab^	6.70 ^ab^	11.17 ^c^	16.44 ^c^
	100	6.70 ^ab^	6.65 ^b^	11.71 ^bc^	16.46 ^bc^
40:60	0	6.70 ^ab^	6.60 ^b^	11.50 ^c^	16.54 ^abc^
	33	6.65 ^bc^	6.75 ^a^	11.94 ^bc^	16.59 ^ab^
	66	6.70 ^ab^	6.70 ^ab^	13.41 ^a^	16.63 ^a^
	100	6.70 ^bc^	6.65 ^b^	13.76 ^a^	16.63 ^a^
	SEM	0.02	0.03	0.33	0.41
Main effects					
	60:40	6.74 ^a^	6.70 ^a^	10.66 ^c^	16.54 ^ab^
R:C Ratio	50:50	6.67 ^b^	6.63 ^b^	11.58 ^b^	16.51 ^b^
	40:60	6.66 ^b^	6.67 ^ab^	12.65 ^a^	16.59 ^a^
YFWBT	0	6.71	6.66	12.06 ^a^	16.58
	33	6.66	6.68	10.36 ^b^	16.53
	66	6.70	6.67	11.81 ^a^	16.55
	100	6.70	6.65	12.30 ^a^	16.53
Significance of main effect and interaction					
R:C Ratio		0.045	0.004	0.003	0.006
YFWBT		0.801	0.050	0.005	0.407
Interaction		0.003	0.003	0.008	0.043

^a–d^ Means in the same column with different lowercase letters differ (*p* < 0.05); YFWBT = yeast-fermented winged bean tuber; SEM = standard error of the mean.

**Table 4 animals-15-02328-t004:** Effects of replacing soybean meal with yeast-fermented winged bean tuber on in vitro dry matter degradability and in vitro organic matter degradability.

R:C Ratio	YFWBT (%)	IVDMD (%)		IVOMD (%)	
		24 h	48 h	24 h	48 h
60:40	0	51.26 ^a^	53.45 ^e^	51.89 ^a^	56.38 ^e^
	33	45.48 ^cd^	57.27 ^c^	46.87 ^cd^	59.65 ^c^
	66	44.60 ^cde^	57.17 ^c^	46.06 ^d^	59.30 ^cd^
	100	40.78 ^f^	54.34 ^de^	43.09 ^ef^	56.99 ^de^
50:50	0	43.59 ^def^	53.36 ^e^	41.87 ^f^	56.40 ^e^
	33	50.18 ^ab^	56.22 ^cd^	51.50 ^a^	59.15 ^cd^
	66	44.85 ^cde^	52.50 ^e^	46.55 ^cd^	55.93 ^e^
	100	41.89 ^ef^	56.76 ^cd^	44.18 ^e^	59.27 ^cd^
40:60	0	51.20 ^a^	56.09 ^cd^	53.34 ^a^	58.83 ^cd^
	33	52.67 ^a^	64.49 ^a^	53.21 ^a^	65.81 ^a^
	66	47.45 ^bc^	60.67 ^b^	49.49 ^b^	63.13 ^b^
	100	45.71 ^cd^	60.26 ^b^	48.17 ^bc^	62.53 ^b^
	SEM	0.94	0.74	0.57	0.71
Main effects					
	60:40	45.53 ^b^	55.55 ^b^	46.98 ^b^	58.08 ^b^
R:C Ratio	50:50	45.13 ^b^	54.71 ^b^	46.02 ^c^	57.69 ^b^
	40:60	49.26 ^a^	60.38 ^a^	51.05 ^a^	62.58 ^a^
YFWBT	0	48.69 ^a^	54.30 ^c^	49.03 ^b^	57.20 ^c^
	33	49.45 ^a^	59.32 ^a^	50.53 ^a^	61.54 ^a^
	66	45.64 ^b^	56.78 ^b^	47.36 ^c^	59.45 ^b^
	100	42.80 ^c^	57.12 ^b^	45.15 ^d^	59.60 ^b^
Significance of main effect and interaction					
R:C Ratio		0.001	0.004	0.008	0.009
YFWBT		0.002	0.007	0.006	0.0051
Interaction		0.001	0.003	0.002	0.010

^a–f^ Means with different letters in a column are significantly different at *p* < 0.05; YFWBT = yeast-fermented winged bean tuber; SEM = standard error of the mean; IVDMD = in vitro dry matter degradability; IVOMD = in vitro organic matter degradability.

**Table 5 animals-15-02328-t005:** Effects of replacing soybean meal with yeast-fermented winged bean tuber on volatile fatty acid (VFA) concentrations.

R:C Ratio	YFWBT (%)	Total VFA, mM		Acetic Acid (%, A)		Propionic Acid (%, P)		Butyric Acid (%)		A:P ratio	
		24 h	48 h	24 h	48 h	24 h	48 h	24 h	48 h	24 h	48 h
60:40	0	70.74	78.34 ^c^	62.13	64.86	22.67	26.00 ^a^	15.18	17.96 ^abcd^	2.74	2.49
	33	72.66	81.26 ^c^	61.72	59.95	22.34	22.98 ^bc^	15.93	16.59 ^de^	2.76	2.61
	66	75.56	85.16 ^cb^	62.95	58.43	21.45	22.10 ^bc^	15.59	15.57 ^e^	2.93	2.65
	100	72.44	80.04 ^c^	62.33	55.82	22.04	21.67 ^c^	15.62	15.61 ^e^	2.82	2.57
50:50	0	80.99	90.41 ^b^	60.64	54.07	22.80	22.64 ^bc^	16.55	16.95 ^cde^	2.65	2.39
	33	86.18	94.60 ^ab^	62.01	50.52	22.29	21.74 ^c^	15.69	16.88 ^cde^	2.78	2.33
	66	88.46	98.88 ^a^	60.31	50.78	22.76	22.24 ^bc^	16.91	16.99 ^cde^	2.64	2.27
	100	81.65	91.07 ^b^	61.34	58.61	22.54	22.80 ^bc^	16.11	16.26 ^e^	2.72	2.57
40:60	0	85.06	97.62 ^a^	58.38	57.96	23.94	24.19 ^abc^	17.66	17.80 ^bcd^	2.44	2.40
	33	89.75	104.31 ^a^	59.26	54.13	23.88	24.59 ^ab^	16.84	18.54 ^ab^	2.48	2.20
	66	85.23	98.79 ^a^	59.17	58.72	23.86	26.22 ^a^	16.96	19.30 ^a^	2.48	2.24
	100	80.74	93.30 ^ab^	57.24	46.97	24.86	22.45 ^bc^	17.88	18.13 ^abc^	2.30	2.09
	SEM	6.52	7.64	0.82	2.93	0.44	0.79	0.52	0.43	0.08	0.10
Main effects											
	60:40	72.85 ^b^	81.20 ^b^	62.28 ^a^	59.77 ^a^	22.13 ^b^	23.19 ^ab^	15.58 ^b^	16.43 ^b^	2.81 ^a^	2.58 ^a^
R:C Ratio	50:50	84.32 ^a^	93.74 ^ab^	61.07 ^a^	53.50 ^b^	22.60 ^b^	22.35 ^b^	16.31 ^b^	16.77 ^b^	2.70 ^a^	2.39 ^b^
	40:60	85.20 ^a^	98.51 ^a^	58.51 ^b^	54.45 ^b^	24.13 ^a^	24.36 ^a^	17.34 ^a^	18.44 ^a^	2.42 ^b^	2.23 ^c^
YFWBT	0	78.93 ^b^	88.79	60.38	58.97	23.14	24.28	16.47	17.57	2.61	2.43
	33	82.86 ^a^	93.39	61.00	54.87	22.89	23.11	16.16	17.34	2.67	2.38
	66	83.08 ^a^	94.28	60.81	55.98	22.69	23.52	16.49	17.28	2.68	2.39
	100	78.28 ^b^	88.14	60.30	53.80	23.15	22.30	16.53	16.67	2.61	2.41
Significance of main effect and interaction											
R:C Ratio		0.001	0.003	0.003	0.021	0.001	0.010	0.004	<0.01	<0.01	<0.01
YFWBT		0.002	0.091	0.691	0.211	0.523	0.050	0.817	0.12	0.55	0.92
Interaction		0.813	0.004	0.463	0.083	0.445	0.004	0.477	<0.05	0.39	0.17

^a–e^ Means with different letters in a column are significantly different at *p* < 0.05; YFWBT = yeast-fermented winged bean tuber; SEM = standard error of the mean.

## Data Availability

The original contributions presented in this study are included in the article; further inquiries can be directed to the corresponding author.

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
