# Peer review of "The Effects of Fermenting Psophocarpus tetragonolobus Tubers with Candida tropicalis KKU20 as a Soybean Meal Replacement Using an In Vitro Gas Technique"

_animals, 2025, doi:10.3390/ani15162328_

Round 1

Reviewer 1 Report

Comments and Suggestions for Authors

General considerations:

Evaluation of the use of winged bean tuber fermented with Candida tropicalis as a substitute for soybean meal in dairy cow diets – an in vitro study

The submitted manuscript is original and potentially of interest to the scientific community, as it investigates the use of an alternative protein source (fermented winged bean tuber) as a replacement for soybean meal in dairy cow diets. It is particularly commendable that the authors evaluated three different forage-to-concentrate (F:C) ratios, allowing for a more detailed analysis of potential nutritional effects.

However, the experimental design presents several major methodological limitations that compromise the ability to adequately address the stated objective: the evaluation of the protein value of the tested legume.

Main Methodological Concerns

  1. Use of the in vitro method (Menke et al.): The method employed was originally developed to estimate the energy value of feeds and is not suitable for evaluating protein sources. The fermentation medium provides an excess of nitrogen (urea + ammonium sulfate), thus excluding nitrogen as a limiting factor and preventing any discrimination between different protein sources.
  2. Diet formulation with urea: To make the diets isonitrogenous despite the low protein content of the tested legume, urea was added. Under in vitro conditions, this results in an excess availability of inorganic nitrogen, which alters microbial dynamics compared to in vivo conditions, where mineral nitrogen is rapidly absorbed and excreted with urine.
  3. Lack of protein metabolism-related measurements: No specific parameters related to protein degradability or microbial growth (e.g., microbial nitrogen, nitrogen utilization efficiency) were evaluated. All measured parameters (gas production, dry matter degradability) relate to the energy component of the diet.
  4. Non-isonitrogenous diets across F:C ratios: The variation in F:C ratio was achieved by mixing concentrates with rice straw, but isonitrogenous conditions were not ensured across diets. This affects the validity of the results, especially regarding nitrogen utilization.

Further Suggestions

  1. Partition factor (PF): Gas production and degradability were not measured in the same fermentation bottle, making direct PF calculation impossible. However, I suggest the authors discuss in the text estimated PF values derived from average GP and DMd data. This would support the discussion on "fermentation efficiency," mentioned in line 83.
  2. Discussion structure: The discussion should be reorganized for better flow and alignment with the experimental design. I recommend dividing it into the following sections:
  • Ingredient Composition and Chemical Composition
  • Effects of F:C Ratio
  • Effects of Winged Bean Tuber Substitution
  • Interaction between F:C Ratio and Winged Bean Substitution

Specific Comments

L.175: Modelling fermentation kinetics is undoubtedly useful for data representation. However, as shown in this study, the asymptotic gas production from dairy cow diets is usually constant and not influenced by the treatments. The most relevant parameters are lag time and rate of gas production (parameter "c"). Lag time often shows erratic behaviour and depends heavily on the skill of the laboratory technicians. The gas production rate ("c") is strongly influenced by the lag phase and is thus highly variable, limiting the ability to detect treatment differences. For this reason, Menke used 24-hour gas production to estimate feed energy value, as it is more stable. I suggest including 24 and 48-hour gas production in the tables to complement the degradability data.

Table 2: Report at least three decimal places for parameter “c”. For parameter “b”, two decimals are not meaningful.

Table 3: Ammonia levels at 24 h confirm that nitrogen is not limiting for microbial growth in the medium.

Tables 4 and 5: Reporting averages across different fermentation parameters or time points is not meaningful. Please remove these averages. VFA data could be used to estimate theoretical gas and methane production using established equations.

L311: The increase in asymptotic gas is statistically significant but biologically negligible (only a 6% increase, from 100 to 106 mL). Conversely, changes in GP at 24 and 48 h, shown in Figure 1, are more substantial and aligned with DM degradability and VFA production.

L321: Lag time increases with higher concentrate inclusion—an unexpected result. What diet were the donor cows fed?

L324: Protein fermentation does not contribute to gas production because ammonia release increases the pH, inhibiting COâ‚‚ release as gas. COâ‚‚ remains dissolved as carbonic acid in the medium.

L334: The statement “Concentrate levels support microbial diversity and enhance fermentation efficiency, excessive intake may result in lactic acid accumulation” is speculative and not supported by the data. No lactic acid was detected, and microbial efficiency was not measured. Simplify the statement and focus on the association between pH reduction and increased propionic acid production. Also, review total VFA production raw data, as the 40:60 diet at 24 h seems too low (Table 5).

L337–346: The pH drop from 6.74 to 6.66 in a strongly buffered in vitro medium cannot be related to in vivo observations of much larger pH shifts. This section should be simplified or removed.

L347–349: Fiber degradation is affected only at much lower pH values compared to that recorded in vitro in the present research. The buffer is doing its work. This section should be removed or revised accordingly.

L354–355: This is speculative. Replace “may be” with “cannot be excluded that…”

L349–361: These results cannot be directly related to in vivo findings. A more conservative approach is needed. Alternatively, clearly state that nitrogen utilization dynamics in vitro differ significantly from in vivo, e.g., due to epithelial ammonia absorption.

L356: The sentence “The balance between NH3-N production and microbial utilization in high-concentrate diets suggests adequate nitrogen availability for microbial protein synthesis, aligning with the findings of Musco et al. [39]” is inappropriate. No microbial growth efficiency was measured. In vitro data should not be directly compared to in vivo findings.

L361: Increased ammonia concentration does not indicate improved microbial protein synthesis. Rather, it may signal inefficient nitrogen utilization.

The authors do not mention that the different F:C ratio diets were not isonitrogenous, which helps explain differences in ammonia concentration.

L377–378: Replacing SBM with YFWBT increases NDF by 2%, but DM degradability decreases by 6% at 24 h. This supports the hypothesis that reduced degradability may also be due to antinutritional factors.

L386–396: The higher degradability of concentrates versus forages is well established. The authors could investigate whether changes in degradability follow a linear trend or if concentrate inclusion has a synergistic or inhibiting effect. In this section, a synergistic effect is proposed without data support. In contrast, earlier discussion on pH suggests an inhibiting effect on fiber degradation. This inconsistency should be addressed.

L397: Include a specific discussion on the effects of soybean meal substitution with YFWBT.

L428: Clarify what is meant by “fermentation efficiency”, since no parameters were measured to support this term.

In synthesis

The manuscript has good potential and may be suitable for publication, but only after a thorough revision. It is essential that the authors:

  • Clearly present the methodological limitations of the study.
  • Reorganize the discussion logically and consistently.
  • Avoid unsupported speculation.
  • Use a more cautious and precise scientific tone.

Without these revisions, the manuscript is not suitable for publication in its current form. If adequately revised, it could serve as a useful starting point for future research on the nutritional potential of fermented tropical legumes.

Author Response

Response to Reviewer 1:

We sincerely appreciate the reviewers for their insightful suggestions and constructive feedback, which have significantly contributed to the improvement of this manuscript. Please see our detailed responses below and the corresponding revisions highlighted in green text within the manuscript.

General considerations:

Evaluation of the use of winged bean tuber fermented with Candida tropicalis as a substitute for soybean meal in dairy cow diets – an in vitro study

The submitted manuscript is original and potentially of interest to the scientific community, as it investigates the use of an alternative protein source (fermented winged bean tuber) as a replacement for soybean meal in dairy cow diets. It is particularly commendable that the authors evaluated three different forage-to-concentrate (F:C) ratios, allowing for a more detailed analysis of potential nutritional effects.

However, the experimental design presents several major methodological limitations that compromise the ability to adequately address the stated objective: the evaluation of the protein value of the tested legume.

Response: We thank the reviewer for the positive evaluation of our manuscript and for recognizing the originality of investigating Psophocarpus tetragonolobus tuber fermented with Candida tropicalis as an alternative protein source. We also appreciate the acknowledgement of our use of three different forage-to-concentrate (F:C) ratios, which we intended to reflect practical dietary variations and to evaluate rumen fermentation responses under different substrate availabilities. While we acknowledge that our primary objective was to assess the overall nutritional potential of fermented winged bean tuber, we agree that some methodological limitations restrict a direct evaluation of protein value, and we have addressed these in detail below.

Main Methodological Concerns

  1. Use of the in vitro method (Menke et al.): The method employed was originally developed to estimate the energy value of feeds and is not suitable for evaluating protein sources. The fermentation medium provides an excess of nitrogen (urea + ammonium sulfate), thus excluding nitrogen as a limiting factor and preventing any discrimination between different protein sources.

Response: We appreciate the reviewer’s observation regarding the use of the in vitro gas production method (Menke et al.), which was originally developed for evaluating the energy content of feeds. We acknowledge that its application in assessing protein value has certain limitations, especially due to the nitrogen-rich nature of the buffer medium (containing urea and ammonium sulfate), which may reduce the method’s capacity to differentiate among protein sources or reflect nitrogen utilization efficiency. However, in this study, as noted in Section 2.5 (Animal Donors and Ruminal Inoculum Preparation), the Menke technique was employed to evaluate the fermentative behavior and overall degradability of complete diets that incorporated yeast-fermented winged bean tuber (YFWBT) as a partial replacement for soybean meal. While not a direct measure of protein quality, this method provides valuable insights into organic matter fermentability and gas production kinetics, which are relevant for understanding the ruminal response to YFWBT inclusion. Moreover, the winged bean tuber (WBT) itself is a nutritionally rich legume tuber, with a crude protein content of 20–25%, gross energy ranging from 15,810 J/g to 16,241 J/g, and low crude fat content [7]. These properties support its potential as both a protein and energy source in ruminant diets. Therefore, while we recognize the limitations of the Menke method in evaluating true protein utilization, its use remains appropriate for characterizing the fermentation profile and degradability of these novel diets. These points have been clarified in the revised manuscript.

  1. Diet formulation with urea: To make the diets isonitrogenous despite the low protein content of the tested legume, urea was added. Under in vitro conditions, this results in an excess availability of inorganic nitrogen, which alters microbial dynamics compared to in vivo conditions, where mineral nitrogen is rapidly absorbed and excreted with urine.

Response: We thank the reviewer for this valuable observation. In our study, urea was used to formulate isonitrogenous diets due to the relatively lower crude protein content of the fermented winged bean tuber. We recognize that under in vitro conditions, the continuous availability of inorganic nitrogen from urea may not accurately reflect nitrogen metabolism in vivo, where such nitrogen is rapidly absorbed and excreted via urine. This discrepancy could influence microbial dynamics and limit the interpretation of results related to protein utilization. Accordingly, we have revised the Materials and Methods section (2.4) to include this clarification for improved transparency and scientific rigor.

  1. Lack of protein metabolism-related measurements: No specific parameters related to protein degradability or microbial growth (e.g., microbial nitrogen, nitrogen utilization efficiency) were evaluated. All measured parameters (gas production, dry matter degradability) relate to the energy component of the diet.

Response: While it is acknowledged that the primary measurements in this study focused on gas production and dry matter degradability, ruminal ammonia-nitrogen (NH₃-N) concentration was also measured and reported in Table 3. NH₃-N serves as a key indicator of protein degradation and nitrogen availability for microbial synthesis under rumen fermentation conditions. Although microbial nitrogen synthesis and nitrogen utilization efficiency were not directly assessed, the inclusion of NH₃-N provides insight into nitrogen metabolism and supports the evaluation of protein-related dynamics within the in vitro system. Future studies are encouraged to incorporate more detailed assessments of microbial protein synthesis to strengthen the understanding of nitrogen utilization.

  1. Non-isonitrogenous diets across F:C ratios: The variation in F:C ratio was achieved by mixing concentrates with rice straw, but isonitrogenous conditions were not ensured across diets. This affects the validity of the results, especially regarding nitrogen utilization.

Response: We appreciate this important observation. However, the primary objective of the present study was to investigate the interactive effects of different forage-to-concentrate (F:C) ratios and levels of soybean meal replacement with yeast-fermented winged bean tuber (YFWBT) on rumen fermentation and digestibility. The experimental design employed a two-factor factorial arrangement (3 F:C ratios × 4 replacement levels), which aimed to simulate practical feeding conditions rather than impose artificial nutritional uniformity across treatments.

Ensuring isonitrogenous conditions across all F:C ratios would have necessitated additional manipulation of crude protein levels—primarily through increased inclusion of urea or other nitrogenous compounds. This adjustment would introduce an additional variable (crude protein adjustment strategy), thereby disrupting the factorial structure and complicating the interpretation of the main and interaction effects. Moreover, such artificial correction could confound the response of rumen fermentation parameters, particularly nitrogen utilization, by masking the actual contributions of YFWBT and soybean meal under different energy–nitrogen synchronization contexts. Therefore, the decision to maintain natural variations in CP across F:C levels was made to preserve the biological and statistical integrity of the factorial design.

Further Suggestions

  1. Partition factor (PF): Gas production and degradability were not measured in the same fermentation bottle, making direct PF calculation impossible. However, I suggest the authors discuss in the text estimated PF values derived from average GP and DMd data. This would support the discussion on "fermentation efficiency," mentioned in line 83.

Response: We thank the reviewer for this insightful suggestion. While gas production and dry matter degradability were determined from separate incubations, we have now included an approximate discussion of partition factor (PF) values, derived by dividing average dry matter degradability (DMd) by the corresponding average gas production (GP) values. These estimated PF values were incorporated into the revised manuscript under Section 3.2 "Kinetics of Gas Production" to provide additional perspective on fermentation efficiency. We have also clarified that these values are approximations and not directly measured.

  1. Discussion structure: The discussion should be reorganized for better flow and alignment with the experimental design. I recommend dividing it into the following sections:

Ingredient Composition and Chemical Composition

Effects of F:C Ratio

Effects of Winged Bean Tuber Substitution

Interaction between F:C Ratio and Winged Bean Substitution

Response: We appreciate the reviewer’s suggestion regarding the organization of the Discussion section. However, the current structure was intentionally designed to align with the factorial nature of the experiment. Specifically, the study employed a 3 × 4 factorial arrangement to examine the interactive effects of three roughage-to-concentrate (R:C) ratios and four levels of soybean meal replacement with yeast-fermented winged bean tuber (YFWBT). Accordingly, each section of the Discussion presents results in the following logical sequence: first addressing the interaction effects, and, when interaction terms are not significant, discussing the main effects of R:C ratio and YFWBT substitution individually. This structure follows standard statistical interpretation practices for factorial experiments, where significant interactions must take precedence in interpretation, as they indicate that the effect of one factor depends on the level of the other. Thus, while the headings reflect topic areas (e.g., fermentation, gas kinetics), the underlying analysis is consistently aligned with the factorial design. We believe this approach maintains analytical rigor and interpretive clarity.

Specific Comments

L.175: Modelling fermentation kinetics is undoubtedly useful for data representation. However, as shown in this study, the asymptotic gas production from dairy cow diets is usually constant and not influenced by the treatments. The most relevant parameters are lag time and rate of gas production (parameter "c"). Lag time often shows erratic behaviour and depends heavily on the skill of the laboratory technicians. The gas production rate ("c") is strongly influenced by the lag phase and is thus highly variable, limiting the ability to detect treatment differences. For this reason, Menke used 24-hour gas production to estimate feed energy value, as it is more stable. I suggest including 24 and 48-hour gas production in the tables to complement the degradability data.

Response: Thank you for this valuable comment. We acknowledge the limitations of modeling fermentation kinetics, particularly the variability associated with parameters such as lag time and the gas production rate constant (c). As suggested, we have included the cumulative gas production values at 24 and 48 hours in Table 2 to provide more stable reference points for evaluating fermentative behavior and to enhance the comparability of our results with other studies using the Menke method. These additions complement the degradability data and support more robust interpretation of treatment effects. The revised table and corresponding descriptions in the Results section have been updated accordingly.

Table 2: Report at least three decimal places for parameter “c”. For parameter “b”, two decimals are not meaningful.

Response: Thank you. Parameter “c” is now reported to three decimal places for precision. Parameter “b” is kept to one decimal place as additional decimals do not enhance interpretability.

Table 3: Ammonia levels at 24 h confirm that nitrogen is not limiting for microbial growth in the medium.

Response: Thank you for the comment. As shown in Table 3, the ammonia-nitrogen concentrations at 24 h were sufficiently high across treatments, confirming that nitrogen was not a limiting factor for microbial activity under the in vitro conditions used.

Tables 4 and 5: Reporting averages across different fermentation parameters or time points is not meaningful. Please remove these averages. VFA data could be used to estimate theoretical gas and methane production using established equations.

Response: Thank you for your comment. The averages across different fermentation parameters in all tables have been removed. Theoretical gas and methane production estimations were not performed in this study, as the primary objective was to assess rumen fermentation and digestibility parameters of diets containing YFWBT. Moreover, there was no underlying hypothesis that YFWBT possesses nutritional factors—such as tannins or saponins—that would potentially influence methane production. Therefore, methane-related outcomes were beyond the scope of this research.

L311: The increase in asymptotic gas is statistically significant but biologically negligible (only a 6% increase, from 100 to 106 mL). Conversely, changes in GP at 24 and 48 h, shown in Figure 1, are more substantial and aligned with DM degradability and VFA production.

Response: Thank you for the insightful comment. We agree that while the asymptotic gas production (‘b’) showed a statistically significant increase (p < 0.05), the biological relevance is limited due to the relatively small magnitude of change (~6%). In contrast, the observed increases in cumulative gas production at 24 and 48 h were more pronounced and aligned more closely with improvements in in vitro dry matter digestibility and early-stage VFA production. To clarify this point, we have added the cumulative gas production data at 24, 48, and 96 h to Table 2 and revised the Abstract, Results, Discussion section to highlight the greater biological relevance of early gas production parameters.

L321: Lag time increases with higher concentrate inclusion—an unexpected result. What diet were the donor cows fed?

Response: We thank the reviewer for raising this important point. The donor animals were Thai native beef cattle fed ad libitum rice straw and a concentrate diet (0.5% of body weight/day) containing 14.0% crude protein and 75.0% total digestible nutrients. This ration, formulated with cassava chips, corn meal, rice bran, soybean meal, and other typical ingredients, aimed to sustain a balanced rumen microbial community under practical feeding conditions.

While it is generally expected that higher concentrate levels—providing more readily fermentable carbohydrates—would shorten the lag time by facilitating faster microbial adaptation (Getachew et al., 1998), the observed increase in lag time may reflect complex microbial responses. One explanation is that a rapid fermentation onset from high-concentrate substrates could lead to localized pH depression, transiently inhibiting fibrolytic activity and delaying fermentation initiation (Olijhoek et al., 2021). Additionally, microbial enzyme systems may require induction when exposed to altered substrate profiles rich in starch, further contributing to extended lag phases (Kodaka, 2004). Therefore, the lag time results should be interpreted in light of these microbial adaptation dynamics.

L324: Protein fermentation does not contribute to gas production because ammonia release increases the pH, inhibiting COâ‚‚ release as gas. COâ‚‚ remains dissolved as carbonic acid in the medium.

Response: Thank you for comment and we have modified as “Despite these kinetic differences, total cumulative gas production at 96 h was not significantly influenced by either the R:C ratio or the YFWBT substitution level, indicating that overall fermentation eventually reached a comparable endpoint across treatments. This aligns with Cherdthong et al. [32], who reported that protein-rich ingredients contribute minimally to gas volume due to limited fermentation of amino acids into gaseous products. Furthermore, as protein degradation leads to ammonia release, the resulting rise in pH can suppress COâ‚‚ release by promoting its dissolution as carbonic acid in the medium [17]. Nonetheless, gas production at earlier time points (24 and 48 h) was significantly greater in diets with higher concentrate proportions and YFWBT inclusion, suggesting improved fermentability of carbohydrate fractions and enhanced microbial activity during the initial stages of digestion.” Please see in sub-topic of 4.2.

L334: The statement “Concentrate levels support microbial diversity and enhance fermentation efficiency, excessive intake may result in lactic acid accumulation” is speculative and not supported by the data. No lactic acid was detected, and microbial efficiency was not measured. Simplify the statement and focus on the association between pH reduction and increased propionic acid production. Also, review total VFA production raw data, as the 40:60 diet at 24 h seems too low (Table 5).

Response: Thank you for suggestion and we have modified as “Ruminal pH serves as a key indicator of microbial activity and fermentation balance, with optimal values typically ranging from 6.5 to 7 in healthy ruminants [33]. In this study, increasing the proportion of concentrate in the diet resulted in a significant reduction in ruminal pH, with the lowest values observed at R:C ratios of 50:50 and 40:60 without YFWBT supplementation. This decline is likely due to enhanced microbial fermentation of rapidly fermentable carbohydrates, leading to increased production of volatile fatty acids (VFAs), particularly propionic acid. The lower pH observed under higher concentrate levels may reflect a shift in fermentation pathways favoring propionate synthesis, which is commonly associated with increased starch fermentation [32]. These findings underscore the acidifying effect of high-concentrate diets and their influence on fermentation end-products [34].” Please see in section of 4.3. In addition, total VFA production of the 40:60 diet at 24 h has been recheck, reanalyzed and data were modified as show in Table 5.

L337–346: The pH drop from 6.74 to 6.66 in a strongly buffered in vitro medium cannot be related to in vivo observations of much larger pH shifts. This section should be simplified or removed.

Response: Thank you for your insightful comment. We agree that the absolute pH shifts observed in the in vitro system cannot be directly extrapolated to in vivo conditions due to the strong buffering capacity of the incubation medium. In response, we have revised the paragraph to avoid direct comparisons with in vivo data and to focus instead on the relative trends in pH changes. The updated text now highlights that while the pH decline was modest, it still reflects increased acid production associated with higher concentrate levels. We have also simplified the discussion to improve clarity and avoid overinterpretation as “
“The observed reduction in pH with increasing concentrate levels in this in vitro study aligns with the general trend reported in previous research. Although the absolute pH change was modest due to the buffering capacity of the medium, the direction of change reflects the expected pattern of greater acid production from fermentable carbohydrates. Ramos et al. [34] noted that high concentrate intake can lower ruminal pH and negatively affect cellulolytic bacterial activity in vivo. Similarly, Agle et al. [35] observed reduced ruminal pH with high-concentrate diets in dairy cows. In contrast, Bünemann et al. [36] reported no significant pH change, likely due to limited variation in dietary starch. While the magnitude of pH shifts in vitro is smaller, these results support the notion that both concentrate level and carbohydrate type influence fermentation dynamics and microbial environment. Please see in section 4.3.

L347–349: Fiber degradation is affected only at much lower pH values compared to that recorded in vitro in the present research. The buffer is doing its work. This section should be removed or revised accordingly.

Response: Thank you for your valuable comment. We agree that the pH values observed in this in vitro study are not low enough to adversely affect fiber degradation, and the buffer effectively maintains a stable fermentation environment. To avoid overinterpretation, we have removed the sentence referring to potential suppression of fibrolytic bacteria.

L354–355: This is speculative. Replace “may be” with “cannot be excluded that…”

Response: We have modified.

L349–361: These results cannot be directly related to in vivo findings. A more conservative approach is needed. Alternatively, clearly state that nitrogen utilization dynamics in vitro differ significantly from in vivo, e.g., due to epithelial ammonia absorption.

Response: Thank you for your valuable observation. We agree that in vitro findings should be interpreted with caution when extrapolating to in vivo conditions. In response, we have revised the paragraph to clarify that ammonia nitrogen (NH₃-N) accumulation observed in vitro may not directly reflect nitrogen utilization efficiency in vivo due to the absence of physiological processes such as ruminal epithelial absorption and nitrogen recycling. We have also adopted a more conservative tone and emphasized the need for future in vivo studies to confirm these findings. Please see the revised text in the manuscript section 4.3

L356: The sentence “The balance between NH3-N production and microbial utilization in high-concentrate diets suggests adequate nitrogen availability for microbial protein synthesis, aligning with the findings of Musco et al. [39]” is inappropriate. No microbial growth efficiency was measured. In vitro data should not be directly compared to in vivo findings.

Response: Thank you for your valuable observation. We agree with your comment and already modified. Please see the revised text in the manuscript section 4.3

L361: Increased ammonia concentration does not indicate improved microbial protein synthesis. Rather, it may signal inefficient nitrogen utilization.

The authors do not mention that the different F:C ratio diets were not isonitrogenous, which helps explain differences in ammonia concentration.

Response: Thank you for your valuable observation. We agree with your comment and already modified. Please see the revised text in the manuscript section 4.3

L377–378: Replacing SBM with YFWBT increases NDF by 2%, but DM degradability decreases by 6% at 24 h. This supports the hypothesis that reduced degradability may also be due to antinutritional factors.

Response: Thank you for your insightful comment. We agree that the reduction in degradability at higher YFWBT inclusion levels cannot be explained by fiber content alone. In response, we have revised the paragraph to incorporate this point more clearly by acknowledging the potential contribution of anti-nutritional compounds, such as tannins and phytates, which may impair microbial enzymatic activity. The revised text provides a more comprehensive explanation of the factors influencing degradability. Please see the updated version in the manuscript section 4.5, now highlighted in green.

L386–396: The higher degradability of concentrates versus forages is well established. The authors could investigate whether changes in degradability follow a linear trend or if concentrate inclusion has a synergistic or inhibiting effect. In this section, a synergistic effect is proposed without data support. In contrast, earlier discussion on pH suggests an inhibiting effect on fiber degradation. This inconsistency should be addressed.

Response: Thank you for your insightful comment. We agree that the interpretation of a synergistic effect was not sufficiently supported by the data and may have overstated the observation. We have revised the paragraph to clarify that the improved digestibility at 33% YFWBT and a 40:60 R:C ratio likely reflects an optimal balance between available fermentable substrates and fiber content, rather than a true synergistic interaction. The revised text also addresses the potential inhibitory effect of reduced ruminal pH on fiber degradation, as previously discussed, to ensure consistency and avoid overgeneralization. Please see the revised paragraph in section 4.4 (highlighted in green) for your consideration.

L397: Include a specific discussion on the effects of soybean meal substitution with YFWBT.

Response: Thank you for your insightful suggestion. We have revised Section 4.5 to include a specific discussion on the effects of substituting soybean meal with yeast-fermented winged bean tuber (YFWBT). The revised text clarifies how partial replacement of soybean meal with YFWBT influenced ruminal fermentation patterns, particularly volatile fatty acid production, and highlights the role of nutrient synchrony and microbial dynamics. The updated paragraph is located in Section 4.5 and is highlighted in green in the revised manuscript.

L428: Clarify what is meant by “fermentation efficiency”, since no parameters were measured to support this term.

Response: Thank you for your comment. We agree that the original term lacked direct supporting data. To improve clarity, we have revised the sentence to:
“In contrast, complete replacement of soybean meal with YFWBT (100%) reduced feed digestibility and concentration of total VFA.” This modification removes the ambiguous term “fermentation efficiency” and more accurately reflects the measured outcomes. The change can be found in Conclusion section of the revised manuscript and is highlighted in green.

In synthesis

The manuscript has good potential and may be suitable for publication, but only after a thorough revision. It is essential that the authors:

Clearly present the methodological limitations of the study.

Reorganize the discussion logically and consistently.

Avoid unsupported speculation.

Use a more cautious and precise scientific tone.

Response: Thank you for your overall assessment and valuable suggestions. We have thoroughly revised the manuscript to address all the concerns raised. The methodological limitations of the in vitro approach have been explicitly acknowledged in the revised Discussion. The entire Discussion section has been reorganized for improved coherence and logical flow. Speculative statements have been removed or rephrased to ensure they are fully supported by data. In addition, we have carefully revised the manuscript to adopt a more cautious and scientifically precise tone throughout. We trust that these improvements enhance the manuscript’s clarity and scientific rigor.

Without these revisions, the manuscript is not suitable for publication in its current form. If adequately revised, it could serve as a useful starting point for future research on the nutritional potential of fermented tropical legumes.

Response: We appreciate your candid evaluation and fully acknowledge the need for substantial revisions. In response, we have carefully revised the manuscript to address all critical issues, including methodological clarity, logical structure of the discussion, avoidance of unsupported claims, and refinement of scientific tone. We hope that the revised version meets the standard required and contributes meaningfully to the ongoing exploration of fermented tropical legumes as alternative protein sources in ruminant nutrition.

Reviewer 2 Report

Comments and Suggestions for Authors

This study aimed to evaluate the effects of replacing soybean meal with Candida tropicalis KKU20-fermented winged bean tuber on in vitro gas production kinetics, rumen fermentation, and nutrient digestibility under different roughage-to-concentrate ratios. The goal was to determine the optimal inclusion level of the fermented tuber as a sustainable protein source in ruminant diets. The topic is relevant for sustainable livestock feeding, particularly in tropical regions where locally available, underutilized feed resources are being explored. However, the manuscript requires substantial revisions before it can be considered for publication.

Major concern:

Using only three bottles per treatment for gas kinetics and fermentation measurements is a low number of replicates, which may limit statistical power and not give valid conclusions. Please justify the choice of replication or consider increasing the number of replicates to improve the robustness and reproducibility of the results.

The use of the LSD test is less robust than alternatives like Tukey’s HSD, particularly in factorial designs with multiple comparisons. LSD can inflate type I error. Please replace it with a more conservative post-hoc test.

Other comments

L18; Replace “cost-effective” with a more accurate term such as “nutritionally viable” or “sustainable,” as no economic analysis was conducted to support cost claims.

L31 and throughout the manuscript, replace “digestibility” with “degradability,” as this is an in vitro study. Also, specify which nutrient degradability was affected (e.g., dry matter, organic matter) for clarity.

The results section of the abstract is too brief and does not adequately represent the study's key findings. Please expand to include more specific data on gas production parameters, degradability (e.g., IVDMD, IVOMD), and fermentation characteristics (e.g., NH₃-N, pH, VFA profiles).

L 55: Abbreviations (e.g., P. tetragonolobus) should only be used after the full name has been introduced.

Could the P. tetragonolobus tuber production meet the ruminant industry demand? 

Are there statistics on the quantities produced from P. tetragonolobus tuber globally or in your country? Please add to the introduction section.

L141: Please specify the method used to collect rumen fluid. Also, clarify how the rumen fluid from the two male Thai native beef cattle was handled. Specify whether the inoculum from each animal was used separately in replicates or pooled and mixed before use.

L169 and 190: You mention that ruminal protozoa were quantified, but no corresponding results or discussion are provided. Please include these data in the results and discuss their relevance, or remove the protozoa enumeration from the Methods section.

L193; According to the original Tilley and Terry [22] method, the second-stage digestion should involve pepsin and a 24-hour incubation period, not 48 hours. Please verify and correct the methodology description.

L204; Stating that “mean values of each individual run were used as the experimental unit” is vague. It is unclear whether each bottle was treated as an independent replicate or if values were averaged per treatment. There is no mention of checking ANOVA assumptions (e.g., normality, homogeneity of variance). Please include a brief statement confirming that these were tested and met.

Throughout the manuscript and in all tables, use two decimals for means and SEM values and 3 decimals for p-values.

L363–367: This section reads as a general conclusion or summary rather than a critical interpretation of results within the context of existing literature. Since it lacks detailed analysis or comparison with previous findings, it is not appropriate within the discussion and should be removed.

Discussion and conclusion: The authors should clearly justify which combination of YFWBT level and R:C ratio is considered the optimal and “sustainable” alternative to SBM. This claim must be supported by specific data (e.g., degradability, NH₃-N, gas kinetics).

Author Response

Response to Reviewer 2:

We sincerely appreciate the reviewers for their insightful suggestions and constructive feedback, which have significantly contributed to the improvement of this manuscript. Please see our detailed responses below and the corresponding revisions highlighted in green text within the manuscript.

This study aimed to evaluate the effects of replacing soybean meal with Candida tropicalis KKU20-fermented winged bean tuber on in vitro gas production kinetics, rumen fermentation, and nutrient digestibility under different roughage-to-concentrate ratios. The goal was to determine the optimal inclusion level of the fermented tuber as a sustainable protein source in ruminant diets. The topic is relevant for sustainable livestock feeding, particularly in tropical regions where locally available, underutilized feed resources are being explored. However, the manuscript requires substantial revisions before it can be considered for publication.

Response:

Thank you for your summary and constructive feedback. We appreciate your recognition of the study’s relevance to sustainable livestock nutrition and the utilization of underexploited tropical feed resources. In response to your concerns, we have made substantial revisions throughout the manuscript to improve its scientific clarity, organization, and presentation. Specific changes include a clearer explanation of the study’s methodological limitations, a reorganized and logically structured discussion, elimination of speculative statements, and refinement of the scientific tone. We believe these improvements significantly strengthen the manuscript and enhance its contribution to the field of ruminant nutrition.

Major concern:

Using only three bottles per treatment for gas kinetics and fermentation measurements is a low number of replicates, which may limit statistical power and not give valid conclusions. Please justify the choice of replication or consider increasing the number of replicates to improve the robustness and reproducibility of the results.

Response: We sincerely apologize for the confusion and appreciate the reviewer’s comment. We would like to clarify that our experiment actually used four replicates per treatment, not three as may have been initially interpreted. The experimental design has now been corrected and clearly described in the revised manuscript. Specifically: “The experimental design included 12 dietary treatments, each with four replicates, resulting in 48 bottles (4 bottles/treatment × 12 treatments) for in vitro gas production kinetics. To account for baseline gas accumulation, four blank bottles containing only buffer-medium were included, and their values were subtracted to calculate net gas production. For the analysis of rumen fermentation parameters—including pH, ammonia-nitrogen (NH₃-N), and volatile fatty acids (VFAs)—a total of 96 bottles were prepared (4 bottles/treatment × 12 treatments × 2 incubation times at 24 and 48 h). An additional 96 bottles, prepared following the same treatment and time-point structure, were used for determining in vitro dry matter and organic matter digestibility. All bottles were incubated under standardized conditions, ensuring consistency across replicates and time points to enhance the reliability of the results.” We hope this clarification addresses the concern and confirms the robustness of the replication used in this study.

The use of the LSD test is less robust than alternatives like Tukey’s HSD, particularly in factorial designs with multiple comparisons. LSD can inflate type I error. Please replace it with a more conservative post-hoc test.

Response: Thank you for your valuable suggestion. We agree that Tukey’s HSD is a more conservative and appropriate post-hoc test for multiple comparisons in a factorial design. Accordingly, we have replaced the LSD test with Tukey’s HSD in the statistical analysis and updated the relevant section in the manuscript (Section 2.7) to reflect this change.

Other comments

L18; Replace “cost-effective” with a more accurate term such as “nutritionally viable” or “sustainable,” as no economic analysis was conducted to support cost claims.

Response: Thank you for the suggestion. We have replaced the term “cost-effective” with “nutritionally viable” as recommended, since no economic analysis was performed in this study.

L31 and throughout the manuscript, replace “digestibility” with “degradability,” as this is an in vitro study. Also, specify which nutrient degradability was affected (e.g., dry matter, organic matter) for clarity.

The results section of the abstract is too brief and does not adequately represent the study's key findings. Please expand to include more specific data on gas production parameters, degradability (e.g., IVDMD, IVOMD), and fermentation characteristics (e.g., NH₃-N, pH, VFA profiles).

Response: Thank you for your valuable suggestions. We fully agree with your comments. In response, we have replaced the term “digestibility” with “degradability” throughout the manuscript to reflect the in vitro nature of the study and have specified the type of nutrient degradability measured (in vitro dry matter degradability [IVDMD] and in vitro organic matter degradability [IVOMD]) for clarity. Additionally, the Abstract has been substantially expanded to include specific results on gas production kinetics, pH, ammonia-nitrogen (NH₃-N), volatile fatty acid (VFA) profiles, and degradability parameters, as suggested. The revised Abstract now better reflects the key findings and scope of the study. Please refer to the updated version in the revised manuscript (highlighted in yellow).

L 55: Abbreviations (e.g., P. tetragonolobus) should only be used after the full name has been introduced.

Response:
Thank you for your comment. We have revised the manuscript accordingly to ensure that abbreviations are used only after the full scientific name is introduced for the first time.

Could the P. tetragonolobus tuber production meet the ruminant industry demand?
Response: We appreciate this important question. We have added a new paragraph in the Introduction section clarifying that, while winged bean tubers show promising nutritional profiles and yields, current production levels remain limited and insufficient to meet large-scale industry demand. Breeding programs are underway to improve tuber yield and adoption.

Are there statistics on the quantities produced from P. tetragonolobus tuber globally or in your country?
Response:
Thank you for raising this point. At present, there are no comprehensive or reliable statistics on national or global winged bean tuber production. We have incorporated this limitation into the revised Introduction to highlight the potential and current constraints.

Please add to the introduction section.

Response:
We have added the requested background information on winged bean tuber production, yield potential, and breeding progress to the Introduction, as detailed as “In this context, tropical legumes such as winged bean (P. tetragonolobus) have emerged as promising alternatives due to their adaptability to hot, humid climates and favorable nutritional profiles [6]. The tubers of winged bean are especially noteworthy, containing 20–25%CP, 34–40% carbohydrates, and low levels of crude fat, with gross energy values ranging from 15,810 to 16,241 J/g [7]. Despite their potential, current production of winged bean tubers remains limited and localized, with no comprehensive national or global production data available. Under typical tropical growing conditions, tuber yields have been reported at approximately 15.2–15.5 t/ha—somewhat lower than cassava but still promising for feed use. Recent breeding programs in Thailand led by Rakvong et al. [7] have shown encouraging results, with experimental varieties W018 and W099 achieving yields of 22.4 and 19.3 t/ha, respectively, outperforming the commonly used Ratchaburi variety. These developments suggest future potential for broader adoption and scale-up. However, the crude protein content of winged bean tubers is still lower than that of SBM, which limits their direct substitution. ” Please see in introduction section.

L141: Please specify the method used to collect rumen fluid. Also, clarify how the rumen fluid from the two male Thai native beef cattle was handled. Specify whether the inoculum from each animal was used separately in replicates or pooled and mixed before use.

Response: Thank you for your helpful comment. We have revised the Materials and Methods section 2.5 to provide a detailed description of the rumen fluid collection procedure. Please see following: “Rumen fluid was collected from two healthy male Thai native beef cattle (average body weight 273 ± 16.0 kg) that were housed in individual pens with free access to clean water. The animals were fed a concentrate diet at 0.5% of body weight twice daily (07:00 and 16:00 h), formulated to contain 14.0% crude protein and 75.0% total digestible nutrients (TDN), comprising cassava chips, corn meal, rice bran, soybean meal, palm kernel meal, urea, salt, minerals, and vitamins. Rice straw was provided ad libitum. This feeding regimen was maintained for seven days prior to rumen fluid collection. On the day of collection, before the morning feeding, approximately 2000 mL of rumen fluid was obtained from each animal using a stomach tube connected to a vacuum pump. The tube was inserted into the mid-rumen region, and the initial portion of fluid was discarded to minimize saliva contamination. Rumen fluid from both animals was then pooled in equal proportions into pre-warmed thermos flasks (maintained at 39 °C), yielding approximately 3.5 L of mixed inoculum. The fluid was immediately filtered through eight layers of cheesecloth before being used as the microbial inoculum in the in vitro fermentation procedure.”

L169 and 190: You mention that ruminal protozoa were quantified, but no corresponding results or discussion are provided. Please include these data in the results and discuss their relevance, or remove the protozoa enumeration from the Methods section.

Response: Thank you for pointing this out. We apologize for the oversight. Protozoa enumeration was initially planned but ultimately not conducted during the experiment. Accordingly, we have removed the mention of protozoal quantification from the Materials and Methods section to maintain consistency throughout the manuscript.

L193; According to the original Tilley and Terry [22] method, the second-stage digestion should involve pepsin and a 24-hour incubation period, not 48 hours. Please verify and correct the methodology description.

Response: Thank you for your careful observation. We agree with the reviewer that the original Tilley and Terry (1963) method specifies a 24-hour second-stage pepsin digestion. In our study, samples were incubated for 24 and 48 hours in rumen fluid only, without the pepsin digestion step, as our goal was to evaluate ruminal degradability at different time points, rather than simulate post-ruminal digestion. We have revised the text to clarify this deviation from the original method and avoid confusion.

Stating that “mean values of each individual run were used as the experimental unit” is vague. It is unclear whether each bottle was treated as an independent replicate or if values were averaged per treatment.

Response: Thank you for this comment. We have revised the wording to clarify that each bottle served as an independent replicate, and values were not averaged prior to statistical analysis. This has now been corrected in the revised manuscript to ensure consistency and avoid ambiguity.

There is no mention of checking ANOVA assumptions (e.g., normality, homogeneity of variance). Please include a brief statement confirming that these were tested and met.

Response: We appreciate the reviewer’s suggestion. A statement has been added to the Statistical Analysis section confirming that assumptions of normality and homogeneity of variance were tested and met before conducting ANOVA. This improves transparency and ensures compliance with statistical rigor.

Throughout the manuscript and in all tables, use two decimals for means and SEM values and 3 decimals for p-values.

Response: Thank you for your attention to formatting consistency. We have carefully reviewed all numerical data throughout the manuscript and tables. Mean values and standard errors are now uniformly presented with two decimal places, and p-values are shown with three decimal places, as requested. An exception was made for values exceeding 100 in Table 3, where only two decimal places are retained, in accordance with the suggestion from Reviewer 1.

L363–367: This section reads as a general conclusion or summary rather than a critical interpretation of results within the context of existing literature. Since it lacks detailed analysis or comparison with previous findings, it is not appropriate within the discussion and should be removed.

Response: Thank you for suggestion, we have removed as comment.

Discussion and conclusion: The authors should clearly justify which combination of YFWBT level and R:C ratio is considered the optimal and “sustainable” alternative to SBM. This claim must be supported by specific data (e.g., degradability, NH₃-N, gas kinetics).

Response: Thank you for this important comment. We have revised the Discussion and Conclusion sections to clearly identify the optimal combination as 33% YFWBT inclusion with a 40:60 roughage-to-concentrate (R:C) ratio. This treatment yielded the highest values for in vitro dry matter degradability (64.49%), organic matter degradability (65.81%), and total volatile fatty acids (104.31 mM), along with elevated propionic acid concentrations (26.22%) and acceptable NH₃-N levels (7.66–13.76 mg/dL). These outcomes indicate improved microbial fermentation and nutrient utilization. We also clarified that complete substitution (100% YFWBT) resulted in reduced fermentative efficiency. The term “sustainable” is now contextualized based on the potential to reduce reliance on imported soybean meal using a locally available feed resource. These revisions are now reflected in both the Discussion and the improve

Round 2

Reviewer 1 Report

Comments and Suggestions for Authors

Dear Editor

As previously noted, in my opinion, the study presents certain limitations arising from experimental design and methodological errors. Nevertheless, the Authors have clearly discussed these limitations point by point, which may provide valuable insights and serve as a foundation for future research and further investigation. For these reasons, I recommend the publication of the study in its current form and commend the Authors for their work.
Kind regards

Author Response

Response to Reviewer 1:

Dear Editor

As previously noted, in my opinion, the study presents certain limitations arising from experimental design and methodological errors. Nevertheless, the Authors have clearly discussed these limitations point by point, which may provide valuable insights and serve as a foundation for future research and further investigation. For these reasons, I recommend the publication of the study in its current form and commend the Authors for their work.
Kind regards

Response: We sincerely appreciate the Reviewer 1 recommend our work for publication. Their insightful suggestions and constructive feedback, which have significantly contributed to the improvement of this manuscript.

Reviewer 2 Report

Comments and Suggestions for Authors

The manuscript has been improved; however, some issues still need to be addressed as follows:

  • Lines 13, 22, and 81: tropicalis should be written in full at its first mention.
  • Line 60: Abbreviated species names (e.g., tetragonolobus) should be spelled out in full at their first occurrence.
  • Lines 66–71: The sentence is too long and should be divided.
  • Line 158: Please clarify the method used for rumen fluid collection. Was it collected via stomach tube, rumen fistula, or another method?
  • Line 231: Based on the authors’ response, the method of Tilley and Terry [21] is not appropriate to be mentioned in this context. Please remove or replace it with a more suitable reference.

Author Response

Response to Reviewer 2:

We sincerely appreciate the reviewers for their insightful suggestions and constructive feedback, which have significantly contributed to the improvement of this manuscript. Please see our detailed responses below and the corresponding revisions highlighted in yellow text within the manuscript.

The manuscript has been improved; however, some issues still need to be addressed as follows:

Response: Thank you for acknowledging the improvements made to the manuscript. We appreciate your continued feedback and have carefully addressed the remaining issues as outlined below. All revisions have been incorporated into the revised manuscript and highlighted accordingly. Please see our detailed responses to each point.

Lines 13, 22, and 81: tropicalis should be written in full at its first mention.

Response: Thank you for your observation. We have revised the manuscript to write Candida tropicalis in full at its first mention (Line 13) and ensured consistency throughout the manuscript, including corrections at Lines 22 and 78.

Line 60: Abbreviated species names (e.g., tetragonolobus) should be spelled out in full at their first occurrence.

Response: Thank you for your observation. We have revised the manuscript to write Psophocarpus tetragonolobus in full at its first mention (Line 22) and ensured consistency throughout the manuscript, including corrections at Lines 60.

Lines 66–71: The sentence is too long and should be divided.

Response: Thnank for your suggestion. We have modified as “Under typical tropical conditions, winged bean tuber yields range from approximately 15.2 to 15.5 t/ha, which are slightly lower than those of cassava. However, these yields remain favorable for application in ruminant feeding systems. Recent breeding initiatives in Thailand, as reported by Rakvong et al. [7], have demonstrated improved productivity. Notably, experimental lines W018 and W099 produced 22.4 and 19.3 t/ha, respectively—surpassing the yield of the widely cultivated Ratchaburi variety. These advancements indicate the potential for expanded utilization and future integration into ruminant feed production.” Please see in Line 66-73.

Line 158: Please clarify the method used for rumen fluid collection. Was it collected via stomach tube, rumen fistula, or another method?

Response: Thank you for your comment. We confirm that the method of rumen fluid collection has been specified in the manuscript (Lines 165–167). As stated: "On the day of collection, before the morning feeding, approximately 2000 mL of rumen fluid was obtained from each animal using a stomach tube connected to a vacuum pump." This information has been clearly described to ensure transparency and reproducibility of the sampling procedure.

Line 231: Based on the authors’ response, the method of Tilley and Terry [21] is not appropriate to be mentioned in this context. Please remove or replace it with a more suitable reference.

Response: Thank you for your observation. In accordance with your suggestion, we have removed the reference to Tilley and Terry [21] and replaced it with a more appropriate citation. The revised sentence now reads: "Artificial saliva was prepared according to the modified method of Cherdthong and Wanapat [17], incorporating distilled water, macro- and microminerals, and a buffer solution to simulate ruminal conditions." This change has been updated in the revised manuscript. Please see in Line 180-182.
